# An Efficient and Secure Big Data Storage in Cloud Environment by Using Triple Data Encryption Standard

Mohan Naik Ramachandra [1], Madala Srinivasa Rao [2], Wen Cheng Lai [3,4,*],
Bidare Divakarachari Parameshachari [5], Jayachandra Ananda Babu [6] and Kivudujogappa Lingappa Hemalatha [7]

1    Department of Electronics and Communication Engineering, Sri Dharmasthala Manjunatheshwara Institute of Technology (Affiliated to Visvesvaraya Technological University), Ujire 574240, India
2    Department of Computer Science and Engineering, PACE Institute of Technology and Sciences, Ongole 523272, India
3    Department of Electronic Engineering, National Yunlin University of Science and Technology, Douliu 640301, Taiwan
4    Bachelor Program in Industrial Projects, National Yunlin University of Science and Technology, Douliu 640301, Taiwan
5    Department of Electronics and Communication Engineering, Nitte Meenakshi Institute of Technology, Bengaluru 560064, India
6    Department of Information Science & Engineering, Malnad College of Engineering, Hassan 573202, India
7    Department of ISE, Sri Krishna Institute of Technology, Bengaluru 560090, India
*    Correspondence: wenlai@yuntech.edu.tw or wenlai@mail.ntust.edu.tw

**Abstract:** In recent decades, big data analysis has become the most important research topic. Hence, big data security offers Cloud application security and monitoring to host highly sensitive data to support Cloud platforms. However, the privacy and security of big data has become an emerging issue that restricts the organization to utilize Cloud services. The existing privacy preserving approaches showed several drawbacks such as a lack of data privacy and accurate data analysis, a lack of efficiency of performance, and completely rely on third party. In order to overcome such an issue, the Triple Data Encryption Standard (TDES) methodology is proposed to provide security for big data in the Cloud environment. The proposed TDES methodology provides a relatively simpler technique by increasing the sizes of keys in Data Encryption Standard (DES) to protect against attacks and defend the privacy of data. The experimental results showed that the proposed TDES method is effective in providing security and privacy to big healthcare data in the Cloud environment. The proposed TDES methodology showed less encryption and decryption time compared to the existing Intelligent Framework for Healthcare Data Security (IFHDS) method.

**Keywords:** big data; Cloud; data encryption standard; security; triple data encryption standard





## 1. Introduction

Enterprises have generated huge amounts of big data due to faster development of Information Technology (IT) and these data need to be efficiently secured, stored and processed [1]. Cloud Computing (CC) is a fast developing business storage computing platform and it includes many advantages such as larger storage, scalability and lower cost [2]. Various enterprise users and individuals have opted to outsource the big data into servers of the Cloud for the purpose of processing and storage. The CC and big data emerged in the 1990s as the earlier stage of technology development in the business. The increase in big data showed the development in volume of data, variety and velocity [3]. The analytics of CC and big data evolved together with the development of information management from basic reporting and querying, advanced analytics, business intelligence and machine learning [4]. The application of CC and big data is on the basis of combining the resources, dynamic on-demand service, capabilities and requirements of integrated services [5]. CC quickly searches for the dynamic and real time information in a shorter

period of time and can transfer the latest data, status of storage and real time information to the Cloud user to obtain the calculation outcome in a shorter time by enhancing the efficiency of economic benefits and distribution [6].

Cloud computing is an emergent technology in data analytics, which is used to retrieve, store and share big data in a distributed environment. Each day individuals and enterprises are storing their data in the server of the Cloud. The authorities of enterprises and individuals are starting to worry about the safety of big data in the Cloud [7]. The Cloud provides three types of services such as software, infrastructure and platform, but delivering the security to big data in the Cloud is the most difficult issue. Generally, the government data, medical data and military data include sensitive details that need to be stored in the environment of the Cloud, but users are not sure about the security given by the service providers [8]. The Cloud includes many advantages but still it lacks the security to store data. It is less popular to store the big data in a single Cloud because of the failure of resource availability and also it includes some conditions where the inside malicious attackers will steal the data from a single Cloud [9]. The existing privacy preserving approaches have several drawbacks such as lack of data privacy, inaccurate data analysis, and completely rely on third parties [10]. Therefore, the privacy of data is another major and important concern in the context of big data. There is a great public fear regarding the inappropriate use of private data, particularly through relating data from multiple sources. Handling privacy is a socio-technical problem, which must be realized to take advantage of big data. To overcome such an issue, Triple Data Encryption Standard (TDES) methodology is proposed in this article to further enhance security in the Cloud environment, especially related to healthcare applications. The contributions of this research are discussed as below:

1.  Implementation of an effective TDES methodology to improve the security in healthcare applications. In this study, the input data are partitioned into three categories based on their importance, and then the encryption technique is applied. The TDES methodology partitions the data for balancing the key length and key strength. Triple encryption is applied for highly sensitive data, double encryption is applied for medium sensitive data, and single encryption is applied for low sensitive data.
2.  Effective handling of symmetric key and data partition helps in better data structuring that improves the network efficiency. The three subkey and key padding techniques further assist in handling the data structure.
3.  The proposed TDES methodology consumes an encryption time of 360 ms; of the existing methods, Tabu search requires 439 ms, MA-ABE technique requires 463 ms, and Hybrid encryption requires 535 ms of encryption time. In addition, the proposed TDES methodology has 98% of packet delivery ratio, while of the existing methods, Tabu has 95%, MA-ABE method has 95%, and Hybrid encryption has 91% of packet delivery ratio. The evaluation measures encryption time and packet delivery ratio show the effectiveness of the proposed TDES methodology.

The paper is organized as follows: the review of existing methods and problem statements are present in Sections 2 and 3, the proposed approach for securing the big healthcare data in Cloud computing by using TDES methodology is explained in Section 4, the experimental results are explained in Sections 5 and 6 describes the conclusion of proposed research.

## 2. Literature Review

Computing infrastructure, especially Cloud computing, plays a significant role in information and knowledge extraction. Efficient handling of big data often poses new security challenges for traditional data encryption standards, methodologies and algorithms. Previous studies of data encryption focused on small-to-medium-size data, which does not work well for big data due to issues in the performance and scalability. Thus, effective policies for data access control and safety management need to be formulated for dealing with big data and this requires incorporating new data management systems. The unprecedented networking among smart devices and computing platforms contributes

to big data but poses privacy concerns, as it involves digitally recording an individual's location, behavior and transactions. The existing techniques on the basis of big data on Cloud computing methods are reviewed in this section.

Zhicheng Zhou et al. [11] developed a Cloud computing method for processing big data and optimizing the performance of multimedia interaction. The radial basis function of neural networks in the Cloud of neural network performs Map-Reduce on Cloud computing. Map reduce and error back propagation algorithm is trained for effective mapping of multi-layer neural networks to process multimedia communication. The developed model has fewer iterations, faster convergence speed and better acceleration speed. The parallelization technique with the developed neural network was not suitable for large scale application of data distribution in computing environments.

Marzieh Mokarram et al. [12] developed a framework by combining Dempster–Shafer theory and Cloud computing for extracting the location to cultivate orange trees. The developed method was applied to obtain the weights for input parameter and Cloud computing was utilized to create the integrated solution that is cost effective by gathering the details from various geographic regions. The interpolation maps for every parameter were evaluated by using inverse distance weighting approach in geographic information system. The developed method successfully predicted the specific location for cultivating the trees by creating various maps. However, when the degree of confidence was increased, the values for land suitability to cultivate trees were decreased, because of the relative humidity in lesser parts that showed the precision value nearer to zero.

Jiaxing Li et al. [13] utilized the technique of blockchain and developed a public auditing approach. Initially, the data owner split the files into blocks and encrypted them to generate the tags of encrypted blocks by using the operation of hash. Then, the data owners established Merkle Hash Tree using hashtags and held the roots for the immediate confirmation of data integrity. Further, the hashtags were forwarded to other data owners which copy every tag stored in the network of blockchain and they uploaded the encrypted blocks to Cloud service provider. The developed method effectively protected against the malicious users and 51% of attacks in the network of blockchain. However, the security and effective services were limited in the developed blockchain based public auditing approach.

Youssef M. Essa et al. [14] developed an intelligent security method known as intelligent framework for the security of healthcare data called as IFHDS. Column based technique is applied large scale data security in developed system that has lesser impact on data processing. The developed technique masks the personal data and encrypt the sensitive data. The sensitive data were split into various parts on the basis of sensitivity levels where every part was separately stored in the distributed Cloud storage. The developed method secured the sensitive data of patients but with a higher computational time. The data owners manage the file configuration of data level security.

Yibin Li et al. [15] applied a distribution storage technique with effective security awareness to secure big data storage distribution in Cloud computing. The developed method utilized algorithms such as alternative data distribution, secure data distribution and efficient data conflation. The developed system divided the files and data were distributed separately in the distributed servers of Cloud. The alternative method was established to identify whether the data packets were split in order to lessen the operational time. The developed security aware method effectively defended the threats of the Clouds and reduced the computational time. However, encrypting all the data using a security algorithm significantly affects the performance requirements in healthcare organizations.

Viswanath and Krishna et al. [16] applied hybrid encryption technique of Advanced Encryption Standard (AES) with s-box and Feistal network to increase the security in the multi-Cloud environment. The hybrid encryption method framework partitions the data and performs the indexing and data encryption. The hybrid encryption framework has a higher performance than AES and Triple DES technique. The hybrid encryption framework technique is able to withstand the Denial of Service (DoS) attack, tampering attack and insider attack. The hybrid encryption framework involves encrypting the

attributes and data, but it requires more computation time. Denis and Madhubala et al. [17] applied two-level Discrete Wavelet Transform steganography and combined it with hybrid encryption technique. The hybrid encryption technique, such as AES and Rivest Shamir Adleman (RSA), is applied for secure diagnosis of data that are embedded with Red Green Blue (RGB) channel in medical cover image. Still, the Adaptive Genetic Algorithm with Optimal Pixel Adjustment Process (AGA-OPAP) needs to enrich the data hiding ability for imperceptibility features.

Ming et al. [18] applied a revocable multi-authority attribute-based encryption (RMA-ABE) for secure Cloud storage. The RMA-ABE technique could withstand adaptive chosen plaintext attack and outperformed Diffie-Hellman problem. The Linear Secret Sharing Scheme (LSSS) technique was applied to boost access policy expressiveness, but it was computationally expensive. Rafique et al. [19] applied CryptDICE to increase the security of the data in the Cloud. The different data encryption techniques were supported by the CryptDICE and annotation was used for access search requirement. The CryptDICE provides appropriate trade-off and execution of encryption decision at various level of data granularity. Jayapandian et al. [20] applied Tabu search concept for the encryption technique and Tabu search technique ensures the reduction of the average encode and decode time in multi-media data. The CryptDICE and Tabu search techniques were effective in scheduling encryption, but they require a local memory table to store the data.

Venkatesan and Chitra [21] applied a security-based distribution storage (SB-DS) technique to reduce the execution time of data partition in the Cloud. The secure data processing, data converging, and data distribution process are present in the model. The fuzzy techniques are applied in this model to partition the sensitive data from normal data. Cuzzocrea et al. [22] applied an attribution-based technique to increases the security of the data in the Cloud. The developed method applied suitable encryption technique for domination relationship. However, the SB-DS technique and the traditional attribution-based technique increases the computation time, while processing the larger databases. Premkamal et al. [23] applied secure deduplication and enhanced attribute-based access control (EABAC-SD) to increase the security of the Cloud data. The EABAC-SD model uses the group key for dynamic ownership management to upload the data and increases the security. However, the EABAC-SD model increases the communication and computation overhead for the encryption and deduplication process. Rashmi et al. [24] applied Improved Chaos Encryption (ICE) method to increase the security based on randomness, but it was a complex process in the large scale databases. The Chaos encryption method is applied with Lorentz 96 technique to improve the security.

In order to address the aforementioned concerns, a new encryption methodology, TDES, is proposed in this research manuscript for efficient and secure big data storage in the Cloud environment with limited computational time.

### 3. Problem Statement

Reviewing the existing studies reveals that most of the encryption techniques increase the computation time, because they require encryption for large scale attributes and data. The IFHDS, AES, genetic algorithm, CryptDICE, SB-DS, ICE, and Tabu search techniques have large key lengths, which is due to the optimization technique becoming trapped into local optima. The existing techniques do not maintain proper structure of data, which leads to higher computation time. To address this concern, a new encryption methodology, TDES, is proposed in this manuscript. The main motivation of this study is to provide efficient and secure big data storage in a Cloud environment with limited computational time.

### 4. Proposed Methodology

In this research, TDES methodology is proposed to provide security for big data in the Cloud environment, especially related to healthcare applications. The initial step in the proposed TDES methodology is data selection or input selection, wherein Healthcare-based big data are considered as an input. After data acquisition, data selection is processed, in

which the encryption of these input data is done by using TDES methodology. The TDES is a standard open encryption methodology, which provides key strength of 112 bits and 168 bits for encryption. The encrypted healthcare-based big data are stored in the Cloud environments. Once the encryption process is completed, after requesting specific data, the decryption process is performed on the corresponding requested data in order to retrieve the data from the Cloud environments. The process of data decryption is accomplished by using TDES methodology. The block diagram of proposed method is shown in Figure 1 and TDES methodology for Healthcare Cloud data is shown in Figure 2.

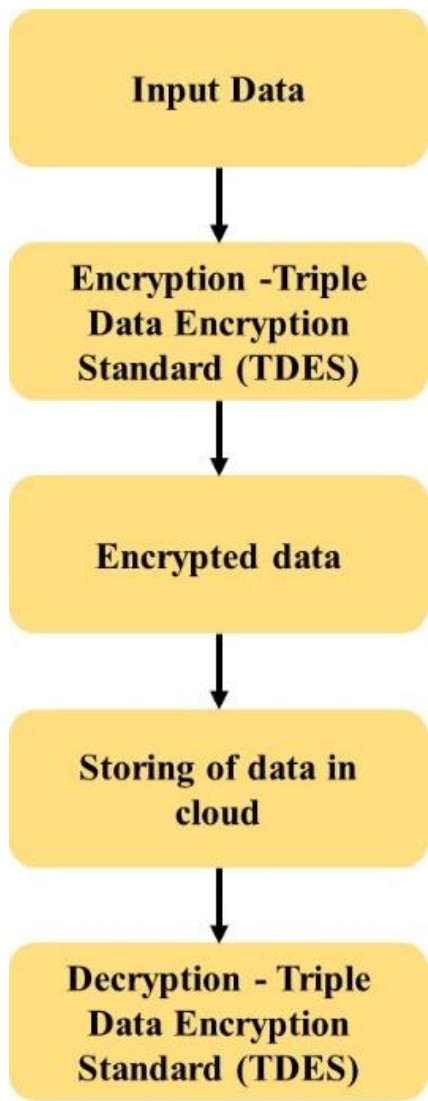

**Figure 1.** The TDES block diagram in encryption.

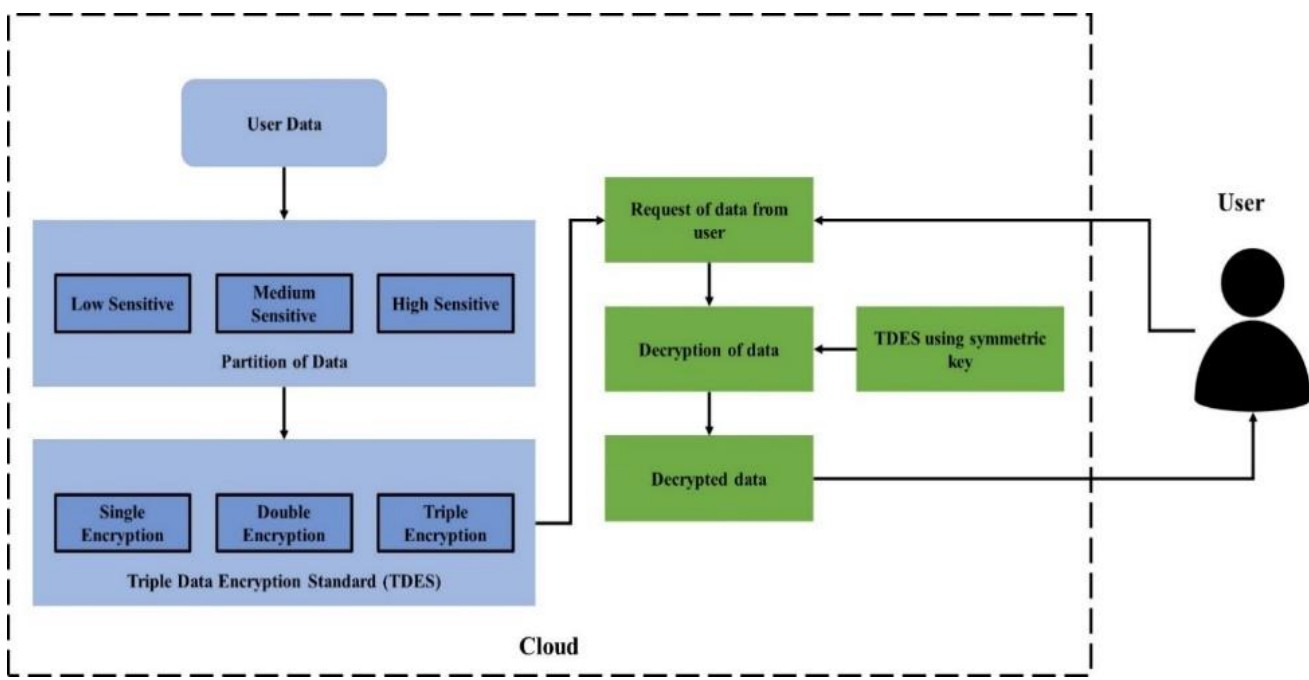

**Figure 2.** Block diagram of proposed TDES for secure big healthcare data storage in Cloud computing.

*4.1. Data Selection*

The initial step in the proposed method is data selection or input selection, where the healthcare dataset is considered as an input. In this manuscript, the real time healthcare dataset is used for experimental investigation. The dataset includes 17 attributes such as patient name, gender, age, month, symptoms, location, disease, resting blood pressure, resting electrocardiographic results, serum cholesterol, maximum heart rate achieved, past history, consultant name, body mass index, body weight, height and job type and has 3024 instances. From the patients' attributes, the required data are selected and considered for the process of encryption. Before performing encryption, the administrator creates a mask for the personal information attributes such as patient name, gender, age, month, location, past history, consultant name, job type and body weight. The example data with personal information attributes and key attributes are mentioned in Tables 1 and 2.

**Table 1.** Example data with personal information.

| Patient Name | Gender | Age | Month | Personal Information | | Consultant Name | Job Type | Body Weight | Patient Mask ID |
| | | | | Location | Past History | | | | |
|---|---|---|---|---|---|---|---|---|---|
| V.J. | M | 47 | June | Jaipur | Y | M.N. | Software engineer | 62 | XXY001 |
| B.U. | M | 21 | September | Chennai | Y | M.I. | Student | 54 | XXY002 |
| M.H. | M | 39 | March | Chennai | N | S.Y. | Bank manager | 78 | XXY003 |
| S.A. | F | 33 | March | Cochin | N | S.Y. | Flim director | 58 | XXY004 |
| V.M. | M | 64 | July | Delhi | Y | N.A. | Farmer | 64 | XXY005 |
| L.H. | M | 36 | May | Mumbai | Y | K.A. | Farmer | 68 | XXY006 |

Male—M, Female—F, Yes—Y, and No—N.

**Table 2.** Example data with key attributes.

| Patient Mask ID | Disease | Symptoms | RBP (mmhg) | Serum Cholesterol (mg/dL) | Heart Rate (BPM) | Height (cm) | BMI | RER (BPM) |
|---|---|---|---|---|---|---|---|---|
| XXY001 | Diabetes | Heartburn | 90 | 182 | 140 | 172 | 21 | 89 |
| XXY002 | Hepatitis | Nausea | 112 | 178 | 117 | 180 | 16.7 | 78 |
| XXY003 | Cardiopathy | Fatigue | 109 | 180 | 133 | 177 | 24.9 | 90 |
| XXY004 | Tuberculosis | Sudden dizziness | 88 | 194 | 122 | 162 | 22.1 | 74 |
| XXY005 | Cardiopathy | Heartburn | 93 | 198 | 167 | 172 | 21.6 | 88 |
| XXY006 | Cardiopathy | Hand pain | 82 | 188 | 138 | 175 | 22.2 | 73 |

Resting blood pressure—RBP, beats per minute—BPM, Body Mass Index—BMI, and resting electrocardiographic results—RER.

The data are considered as an assortment of characters that are interpreted for the purpose of analysis. The expression of input data is explained in Equation (1)

$$S_i = S_1, S_2, S_3, \ldots\ldots, S_n \tag{1}$$

where $S_i$ is the number of inputs selected; $S_n$ is the $n$ number of input fields in selection.

*4.2. Data Encryption*

After selecting the data, next process is input encryption by using TDES methodology. The TDES is a standard open encryption methodology that provides 112 bits and strength of 168 bits for encryption. The TDES utilizes a symmetric key methodology to transmit, read and write the data. The TDES methodology is the effective one, as it utilizes symmetric generation of keys and delivers better security. After the process of encryption, the data are stored in big data. The TDES methodology is generally defined in Equation (2).

$$E^1 = E^3 = E, E^2 = D \tag{2}$$

where $E$ is the single encryption function in TDES, $E^2$ is the double encryption function in TDES, $E^3$ is the triple encryption function in TDES and $D$ is the counterpart of decryption.

Triple Data Encryption Standard (TDES) Methodology

The TDES is a beneficial methodology for encryption as well as decryption because it includes larger sized key lengths, which are longer than many key lengths used in the existing encryption models. The advanced encryption standard is replaced by Data Encryption Standard (DES); it is considered as outdated at present [25]. It is obtained by using single DES thrice and utilizes three subkeys and padding of keys. The keys are augmented to the length of 64 bits, which is recognized for compatibility and flexibility, then converted to inclusion of TDES. There are several types of TDES encryption that are normally recognized in three ways [25], which are explained in the following:

- DES-3EES: The three data encryption standard using three types of different key.
- DES-EDES: The different keys for every three operation of data encryption standard such ad encryption, decryption and encryption.
- DES-EEE2 and DES-EDES2: The different keys for secondary decryption operation.

The $E_K(I)$ and $D_k(I)$ is the encryption and decryption of data encryption standard of $I$ by using $K$ key of DES accordingly. The TDES encryption and decryption operations stands as a compound operation of DES for encryption and decryption, the operations are utilized as below explained.

- Encryption operation of TDES: The $I$ block of 64 bits is converted to block 0 of 64 bits, which is as shown in Equation (3):

$$O = E_{K_3}(D_{K2}(E_{k1}(I))) \tag{3}$$

- Decryption operation of TDES: The $I$ block of 64 bits is converted to block 0 of 64 bits which is as shown in Equation (4):

$$O = D_{K1}\left(E_{K_2}(D_{K3}(I))\right) \tag{4}$$

There are three ways to utilize the combination of subkeys that are considered as standard in the process of encryption and decryption by using TDES which are explained as follows:

- Key option 1: The three subkeys such as $K_1$, $K_2$, $K_3$ are considered as the independent keys, which includes different combination such as 3K3DES.
- Key option 2: The three subkeys such as $K_1$, $K_2$, $K_3$ are considered, where the $K_1$ *and* $K_2$ will have independent or different combination and $K_1$ and $K_3$ will include similar combination such as (2K3DES)
- Key option 3: The three subkeys such as $K_1$, $K_2$, $K_3$ are considered, where all the keys are of similar combination, as shown in Equation (5):

$$K_1 = K_2 = K_3 \tag{5}$$

In the three key options to utilize the subkeys, the first option is best because all the three subkeys includes different combination having effective key length of 168 bits. Therefore, it is difficult to encrypt the data using first option and it is difficult to resolve the usage of the second and third option. The second option includes effective key length of 112 bits where the initial subkey includes similar combination as third subkey. However, the second option is better than utilizing the twice DES encryption. The third option is weak as compared with the first and second option where the first subkey and second subkey are neglected in the process of encryption and decryption, so the key used in the third option is effective, with 56 bits length as similar as the key length utilized in DES cryptographic models.

The TDES operation stands effectively with the counterpart of single DES by using well matched keying options of TDES operations, such as:

- The encrypted plaintexts are calculated by using single DES by decrypting properly by means of effective TDES operation.
- The encrypted plaintexts are calculated by using single TDES by decrypting properly by means of effective DES operation.

### 4.3. Big Data Storage in Cloud

The encrypted big health care data are stored in the Cloud environments, where it effectively supports read–write operations in storage [25,26]. Generally, the structural design of big data includes redundant and scalable supplies of direct attached storage pools clustered or scale-out network attached storage, or the infrastructure will be centered as a storage of object format [27–30]. The infrastructure of big data storage is linked to Cloud computing server nodes that allow quicker processing and retrieval of big data measures. The spark over the Hadoop distributed environment is utilized, which provides memories in Cloud and encryption key distribution to worker machines easily. The key management system is improved for data encryption on column instead of row. The keys are handled by key management system in Hadoop environment. The master nodes are generated for keys at every security level, and to store the keys in key management system to utilize when the queries of end users are stored. In the TDES methodology, the queries of clients by distribution of required keys are related to the workers' queries in the Cloud. All data are running in Yet Another Resource Negotiator (YARN) federations so every Cloud environment utilizes its own resource manager which maintains data processing in a similar Cloud of spark [31,32]. The single YARN data scales use this technique to many nodes of sub-data YARN federating. From the view of application, the federation combines sub-data together as single larger YARN cluster to an application.

### 4.4. Data Decryption

In the data decryption process, the cloud environment data are retrieved by using TDES methodology. The TDES methodology enables the data owner to retrieve data from the Cloud of healthcare organizations. The Hadoop cluster resources manager provides access to data owners to retrieve the data [33,34]. The resource manager will have the access to map the tables of huge data among various data nodes or providers. Every attribute on security level basis uses appropriate decryption key is used to decrypt the data in TDES methodology. The workers in YARN are able to utilize Hadoop decryption in key management system. The resource manager provides many locations in various data nodes once the client requests data from Hadoop or Cloud providers [35,36]. Then, the data nodes send data to TDES methodology and TDES reads the security level of every attribute by using similar methodology. After TDES completes its process, the Hadoop decrypts the data in the directory of output that is configured by users.

### 5. Results and Discussion

The Cloud includes many advantages but still it has limitations in providing the security for storage of big data. Various existing methods were developed to provide the security for big data storage in the Cloud. The existing privacy preserving approaches showed several drawbacks such as a lack of data privacy and accurate data analysis, a lack of efficiency of performance and reliance on a third party, which considers the vulnerability of security and reveals the private data of the individual. The TDES methodology is proposed to improve security of medical data in Cloud computing using Python software on a system with 6 GB GPU, 16 GB RAM, i7 core processor and Windows 10 OS environment. The TDES methodology evaluation uses performance measures as explained in this section. The comparison of the proposed method with the existing methods is described in this section.

### 5.1. Performance Measures

The proposed TDES methodology for big health care data storage in Cloud environment is evaluated using the following metrics. The few security measures that can be used to improve the Cloud computing environment are described below.

- Execution Time

Execution time is also called as CPU time of a particular task is defined as time taken by the system to complete or execute the task; it is defined in Equation (6).

$$Execution\ Time = I * CPI * T \tag{6}$$

where $I$ is the number of instructions in the program, $CPI$ is the average cycles per instruction and $T$ is the time of clock cycle.

- Network Usage

The network utilization is the overall percentage of network bandwidth that is consumed by the traffic of network. The network utilization indicates the point at which the network gets slowdown and shows a need for changes. It is calculated by using Equation (7).

$$Network\ Usage = \frac{Network\ Bnadwidth}{Network\ Traffic} \times 100 \tag{7}$$

- CPU utilization

Central Processing Unit utilization is the overall work handled by CPU and it is utilized to evaluate performance of system. It is calculated by using Equation (8).

$$CPU\ utilization = (100\% - Time\ spent\ in\ an\ idle\ task) \tag{8}$$

### 5.2. Comparative Analysis

The values obtained for the proposed TDES methodology to provide security for the big medical data in Cloud computing and existing methods are explained in this section. The comparison among proposed TDES methodology and existing methods such as IFHDS [14] and SA_EDS_AES [15] in terms of performance measures is given in Table 3.

**Table 3.** Comparison table of proposed TDES methodology with existing methods.

| Methods | Execution Time (Min) | Network Utilization (GB) | CPU Usage (%) |
| --- | --- | --- | --- |
| IFHDS | 53 | 80 | 90 |
| SA_EDS_AES | 55 | 83 | 70 |
| TDES | 48 | 85 | 93 |

Table 3 shows the performance of the proposed method and existing methods in terms of execution time, network utilization and CPU usage. The time taken for encryption, transmission and decryption in medical data is execution time in TDES methodology. The network utilization for the TDES methodology to encrypt, decrypt and transmit the data is estimated. The end-to-end execution utilizes the CPU to deliver better performance and to meet the requirements of security. The proposed TDES methodology has a reduced execution time of 48 min for encryption, storage and decryption as compared to existing methods. The network utilization of proposed method is 85 GB where existing methods used 80 GB and 83 GB. The CPU utilization of to execute the task of proposed TDES methodology is 93%. The proposed TDES methodology provides a simpler technique by increasing the sizes of keys in DES to protect against the attacks and defends the privacy of data. The plotted graph of proposed TDES methodology with the existing method is shown in Figure 3.

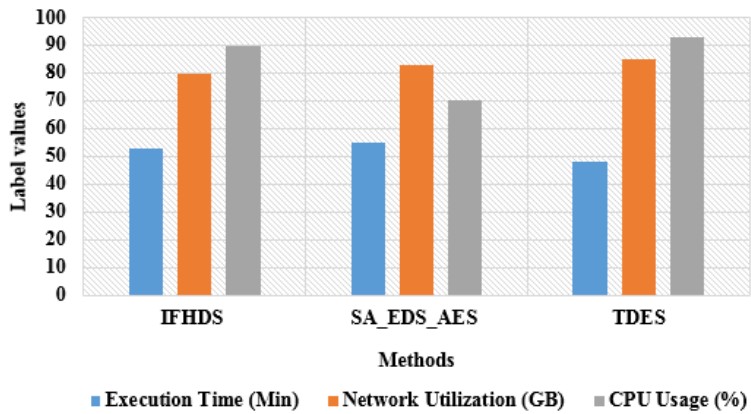

**Figure 3.** TDES methodology is compared with existing techniques.

Table 4 shows the execution time of the proposed TDES methodology for 100 to 500 data size as compared with the existing methods. The execution time will be based on security level of methodology. The proposed TDES has a lower execution time and shows effective encryption, decryption and storage performance. The execution time of proposed TDES methodology for the data size of 500 is 55 min and it provides effective security to the model. The existing IFHDS [14] and SA_EDS_AES [15] consumed more execution time and did not meet all the security requirements. The graphical representation of proposed TDES methodology with existing method in terms of execution time is shown in Figure 4.

**Table 4.** TDES methodology is compared with existing techniques in terms of execution time.

| Data Size | IFHDS (Min) | SA_EDS_AES (Min) | TDES (Min) |
|---|---|---|---|
| 100 | 28 | 25 | 20 |
| 200 | 30 | 30 | 25 |
| 300 | 35 | 35 | 30 |
| 400 | 48 | 43 | 40 |
| 500 | 68 | 60 | 55 |

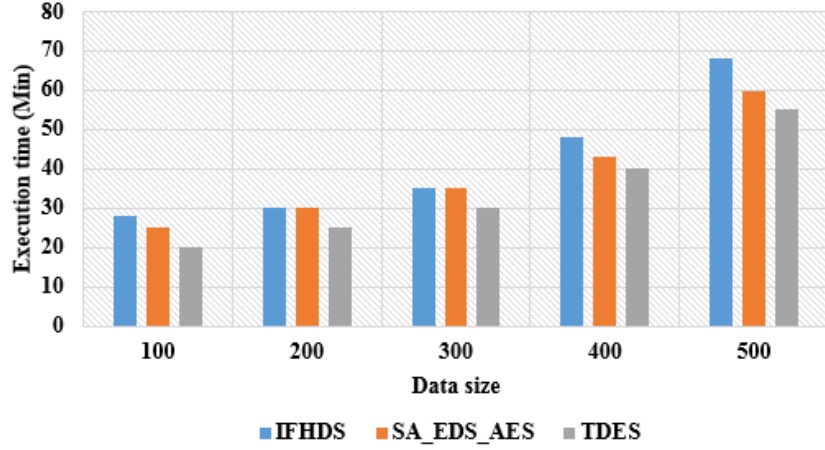

**Figure 4.** TDES methodology in comparison with existing methods in terms of execution time.

The Table 5 shows the network usage of proposed TDES methodology for 100 to 500 data sizes as compared with the existing methods. The network utilization for the TDES methodology to encrypt, decrypt and transmit the data is estimated. The network utilization of proposed method is higher compared to existing methods. The network utilization is the amount of data transmitted among the workers over the clusters during the process of encryption and decryption. The proposed TDES methodology utilizes the network of 0.55 GB for the 500 data size. The proposed TDES methodology provides a simpler technique by increasing the sizes of keys in DES to protect against the attacks and defends the privacy of data. The graphical representation of proposed TDES methodology with existing method in terms of network usage is shown in Figure 5.

**Table 5.** Network usage of TDES is compared with existing techniques.

| Data Size | IFHDS (GB) | SA_EDS_AES (GB) | TDES (GB) |
|---|---|---|---|
| 100 | 0.19 | 0.18 | 0.21 |
| 200 | 0.26 | 0.24 | 0.29 |
| 300 | 0.33 | 0.32 | 0.36 |
| 400 | 0.38 | 0.35 | 0.40 |
| 500 | 0.51 | 0.49 | 0.55 |

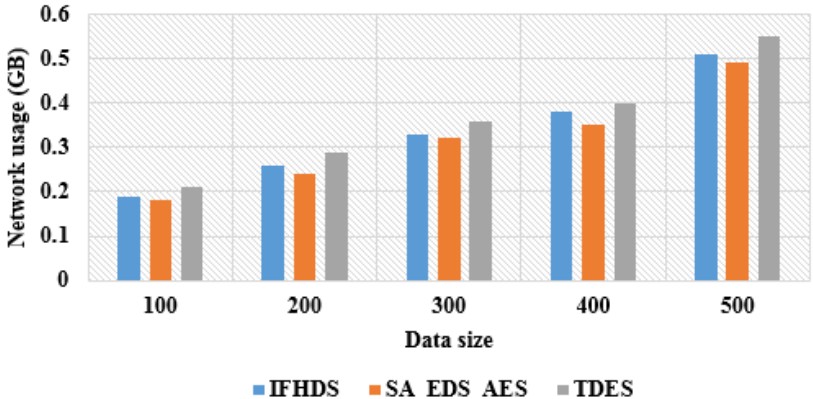

**Figure 5.** TDES network usage is compared with existing techniques.

The comparison table of the proposed TDES methodology with existing methods in terms of CPU utilization is shown. Table 6 shows the CPU utilization of the proposed TDES methodology for 100 to 500 data sizes compared with the existing methods. The end-to-end execution utilizes the CPU to deliver better performance and to meet the requirements of security. The proposed TDES methodology utilizes the CPU of 33% for 500 data size, whereas the existing IFHDS [14] method utilized CPU of 35% and SA_EDS_AES [15] of 34%. The proposed TDES methodology provides a simpler technique by increasing the sizes of keys in DES to protect against the attacks and defends the privacy of data. The graphical representation of the proposed TDES methodology compared with existing methods in terms of CPU utilization is shown in Figure 6.

**Table 6.** TDES CPU usage comparison with existing techniques.

| Data Size | IFHDS (%) | SA_EDS_AES (%) | TDES (%) |
|:---:|:---:|:---:|:---:|
| 100 | 25 | 25 | 24 |
| 200 | 25 | 25 | 24 |
| 300 | 28 | 26 | 28 |
| 400 | 32 | 31 | 30 |
| 500 | 35 | 34 | 33 |

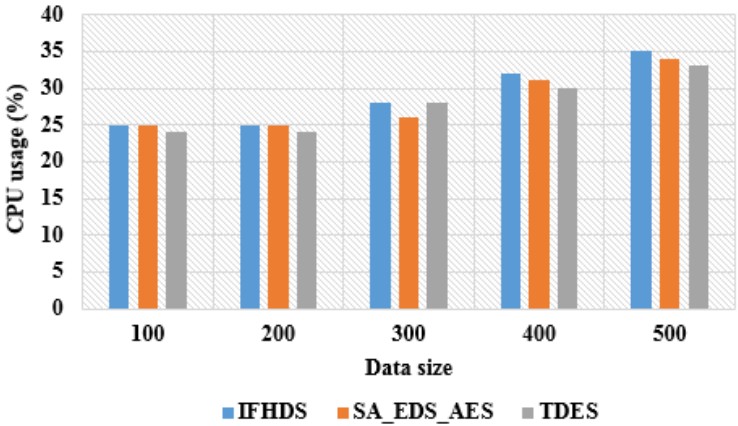

**Figure 6.** TDES CPU usage comparison with existing techniques.

The TDES methodology and existing techniques were implemented with the same data and in the same environment to test their performance. Some recent existing methods such as Hybrid Encryption [16], MA-ABE [18], and Tabu [20] were compared with TDES methodology in the network.

The encryption time of TDES and existing methodologies for various number of blocks were measured and compared, as shown in Table 7 and Figure 7. The number of blocks varied from 100 to 800 to evaluate the TDES and existing methodologies' performance. The TDES methodology has the advantage of balancing the key length and strength of the key based on the importance of data. This methodology helps to reduce the encryption time of the data compared to existing techniques. The symmetric key is applied to handle the structure of data, which supports reading, storing and editing. The proper structure of data handling further reduces the encryption time in TDES methodology. The Hybrid encryption [16] and MA-ABE [18] techniques require encryption for attributes and data that increases the encryption time of the model. The Tabu search techniques has local optima that provides more key length and increases the encryption time. The TDES methodology requires encryption time of 360 ms, Tabu search [20] requires 439 ms encryption time, MA-ABE [18] technique requires 463 ms encryption time and Hybrid encryption requires 535 ms encryption time.

**Table 7.** Encryption time of TDES and comparison with existing techniques.

| Number of Block | Hybrid Encryption (ms) | MA-ABE (ms) | Tabu (ms) | TDES (ms) |
|---|---|---|---|---|
| 0 | 0 | 0 | 0 | 0 |
| 100 | 484 | 400 | 394 | 204 |
| 200 | 486 | 405 | 398 | 260 |
| 300 | 487 | 418 | 399 | 266 |
| 400 | 496 | 433 | 403 | 277 |
| 500 | 521 | 438 | 405 | 327 |
| 600 | 521 | 439 | 411 | 328 |
| 700 | 529 | 442 | 430 | 334 |
| 800 | 535 | 463 | 439 | 360 |

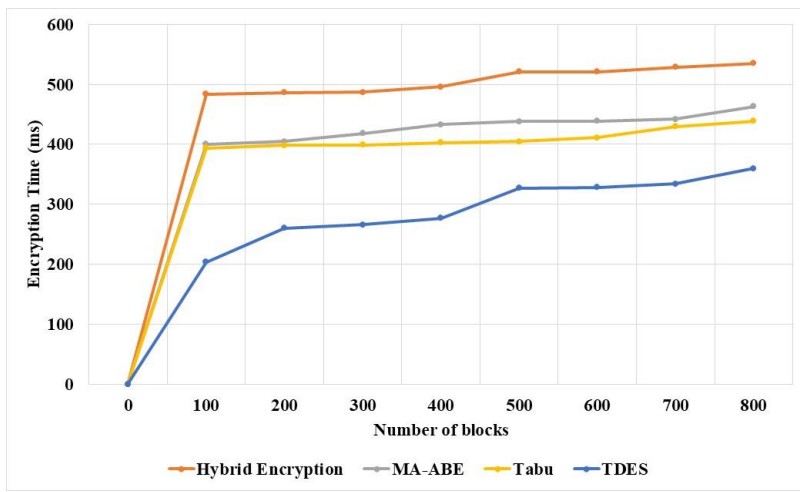

**Figure 7.** TDES and existing techniques encryption time for various numbers of blocks.

Decryption time of the TDES and existing techniques was measured and compared for various numbers of blocks, as shown in Table 8 and Figure 8. The TDES shows significant reduction of decryption time compared to existing techniques in Cloud data security. The TDES methodology has the advantages of handling structured data using symmetric key and balance key length–strength, based on the importance of data. The decryption time is significantly reduced due to key and structure handling. The triple encryption is applied to sensitive data and increases the strength of encryption. The Tabu search techniques [20] uses large key strength to encrypt the data due to local optima in search process. The MA-ABE [18] and Hybrid encryption [16] techniques require decrypting the attribute

and data, which increases the decryption time. The TDES methodology requires 475 ms decryption time, Tabu search [20] requires 603 ms, MA-ABE [18] requires 740 ms and Hybrid encryption [16] requires 793 ms decryption time for 800 blocks of data.

**Table 8.** Decryption time of TDES and comparison with existing methodologies.

| Number of Block | Hybrid Encryption (ms) | MA-ABE (ms) | Tabu (ms) | TDES (ms) |
|---|---|---|---|---|
| 0 | 0 | 0 | 0 | 0 |
| 100 | 767 | 673 | 633 | 428 |
| 200 | 767 | 739 | 673 | 472 |
| 300 | 776 | 730 | 606 | 415 |
| 400 | 783 | 700 | 602 | 560 |
| 500 | 785 | 671 | 657 | 425 |
| 600 | 789 | 679 | 697 | 457 |
| 700 | 792 | 701 | 657 | 522 |
| 800 | 793 | 740 | 603 | 475 |

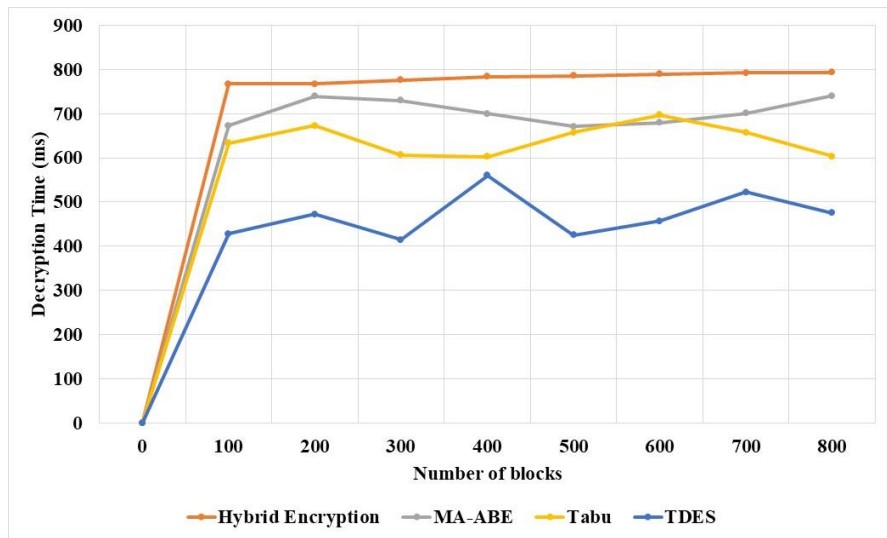

**Figure 8.** TDES and existing techniques decryption time for various number of blocks.

The running time of TDES and existing techniques was measured and compared for various numbers of blocks, as shown in Table 9 and Figure 9. The TDES running time is less compared to existing techniques, due to its efficiency in data structure handling and balance of key length–strength. The TDES methodology partitions the data into three categories based on importance: high sensitive data are applied with triple encryption, medium sensitive data are applied with double encryption, and low sensitive data are applied with single encryption. This methodology helps to reduce total running time of the model compared to existing techniques. The Hybrid encryption [16] and MA-ABE [18] technique have more running time due to encrypting and decrypting both attributes and data. The Tabu [20] search has a high running time and high key strength due to local optima in the search process.

**Table 9.** Running time of TDES and comparison with existing technique.

| Number of Block | Hybrid Encryption (ms) | MA-ABE (ms) | Tabu (ms) | TDES (ms) |
|---|---|---|---|---|
| 0 | 0 | 0 | 0 | 0 |
| 100 | 1393 | 1227 | 971 | 833 |
| 200 | 1348 | 1285 | 1026 | 832 |
| 300 | 1344 | 1112 | 1023 | 816 |
| 400 | 1276 | 1113 | 979 | 985 |
| 500 | 1272 | 1197 | 1043 | 887 |
| 600 | 1267 | 1115 | 954 | 997 |
| 700 | 1240 | 1286 | 902 | 946 |
| 800 | 1239 | 1247 | 1016 | 911 |

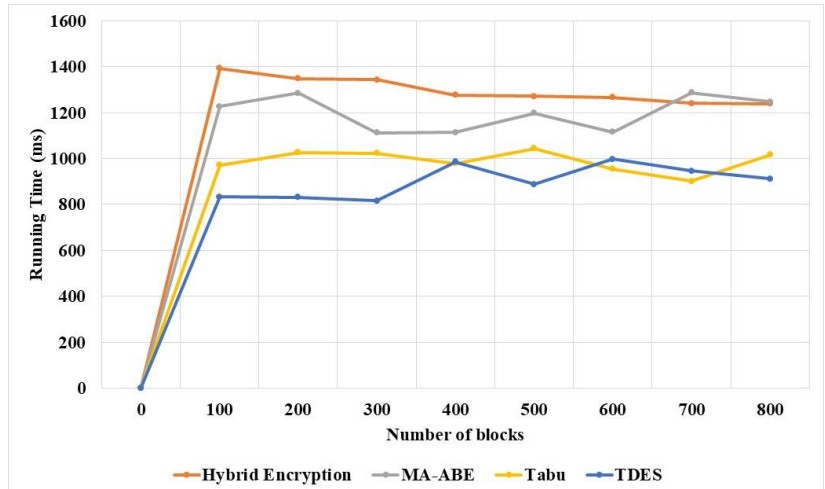

**Figure 9.** TDES and existing techniques running time for various number of blocks.

Latency of TDES and existing techniques were measured and compared for various numbers of blocks, as shown in Table 10 and Figure 10. The TDES has significantly lower latency in the model compared to existing techniques, due to its proper structure of data using symmetric key. The Tabu [20], MA-ABE [18] and Hybrid encryption [16] do not maintain proper structure of data, which involves increases in the latency of the model. The TDES methodology has 78 ms latency, Tabu [20] has 119 ms, MA-ABE has 140 ms and Hybrid encryption has 144 ms latency for 800 number of blocks.

**Table 10.** Latency of TDES and comparison with existing technique.

| Number of Block | Hybrid Encryption (ms) | MA-ABE (ms) | Tabu (ms) | TDES (ms) |
|---|---|---|---|---|
| 0 | 0 | 0 | 0 | 0 |
| 100 | 84 | 71 | 54 | 14 |
| 200 | 84 | 78 | 82 | 45 |
| 300 | 103 | 106 | 89 | 56 |
| 400 | 106 | 118 | 91 | 58 |
| 500 | 116 | 122 | 94 | 61 |
| 600 | 117 | 131 | 99 | 66 |
| 700 | 134 | 135 | 103 | 72 |
| 800 | 144 | 140 | 119 | 78 |

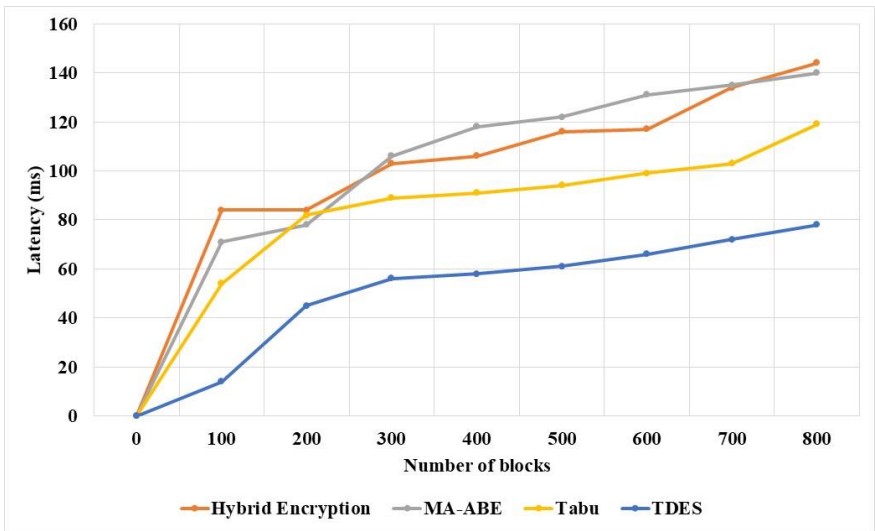

**Figure 10.** Comparing TDES and existing techniques' latency for various numbers of blocks.

The TDES and existing techniques throughput were measured and compared in Table 11 and Figure 11. The TDES methodology supports higher throughput than existing techniques due to its efficiency in structure data handling using symmetric key. The TDES methodology partition the data into three categories based on data importance. The TDES methodology transfers the low sensitive data in high speed due to its single encryption. The Tabu [20], MA-ABE [18] and Hybrid encryption [16] methods do not maintain proper structure and this reduces the throughput of the network. The TDES has throughput of 120 MBPS, and existing Tabu [20] has 106 MBPS, MA-ABE [18] has 68 MBPS throughput, and Hybrid encryption [16] has 60 MBPS throughput.

**Table 11.** Throughput of TDES in comparison with existing techniques.

| Running Time (ms) | Hybrid Encryption (MBPS) | MA-ABE (MBPS) | Tabu (MBPS) | TDES (MBPS) |
|---|---|---|---|---|
| 0 | 0 | 0 | 0 | 0 |
| 100 | 15 | 40 | 71 | 96 |
| 200 | 17 | 44 | 72 | 99 |
| 300 | 24 | 46 | 80 | 100 |
| 400 | 32 | 54 | 81 | 100 |
| 500 | 50 | 59 | 82 | 106 |
| 600 | 52 | 63 | 87 | 113 |
| 700 | 52 | 65 | 104 | 120 |
| 800 | 60 | 68 | 106 | 120 |

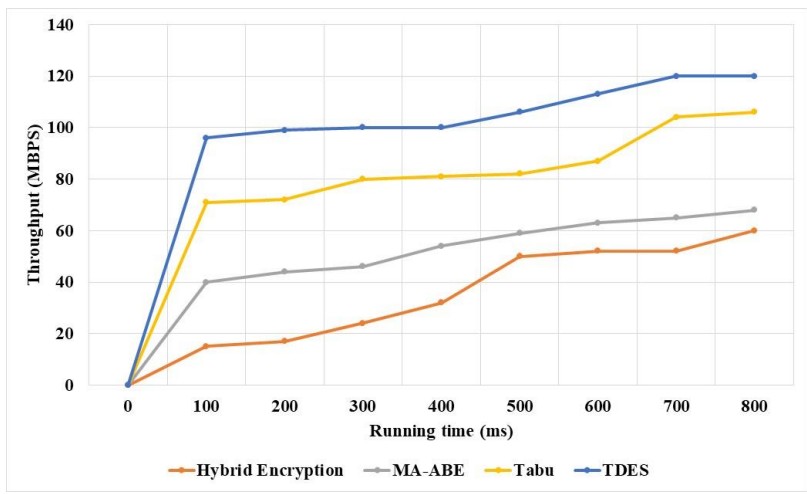

**Figure 11.** Throughput of TDES and existing techniques for various running times.

The packet delivery ratio of TDES and existing techniques were measured and compared for various numbers of blocks, as shown in Table 12 and Figure 12. The TDES methodology has higher packet delivery ratio due to the partitioning of data and structure of handling. The Tabu [20], MA-ABE [18] and Hybrid encryption [16] methods do not maintain proper structure and this reduces the packet delivery ratio in the network. The TDES methodology has 98% packet delivery ratio, Tabu [20] has 95%, MA-ABE method has 95% and Hybrid encryption has 91% packet delivery ratio.

**Table 12.** Packet delivery ratio of TDES and comparison with existing techniques.

| Number of Block | Hybrid Encryption (%) | MA-ABE (%) | Tabu (%) | TDES (%) |
|---|---|---|---|---|
| 0 | 0 | 0 | 0 | 0 |
| 100 | 88 | 86 | 88 | 91 |
| 200 | 88 | 87 | 89 | 91 |
| 300 | 89 | 88 | 90 | 93 |
| 400 | 89 | 90 | 90 | 93 |
| 500 | 90 | 92 | 92 | 93 |
| 600 | 91 | 93 | 92 | 94 |
| 700 | 91 | 95 | 94 | 97 |
| 800 | 91 | 95 | 95 | 98 |

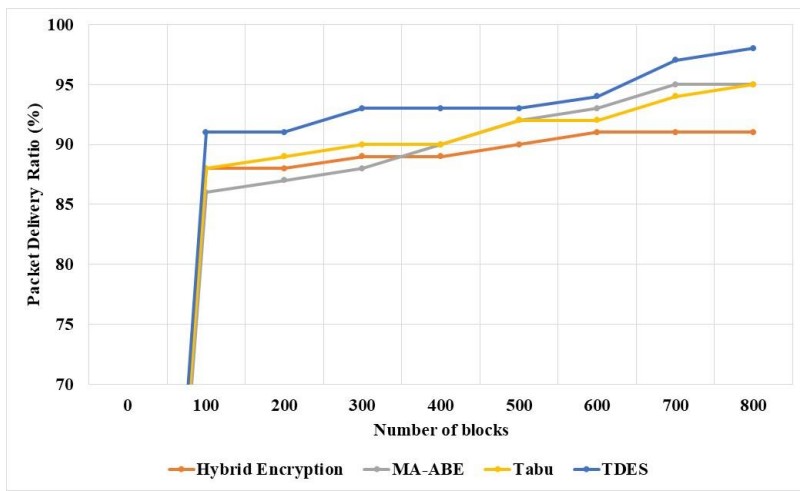

**Figure 12.** Packet delivery ratio of TDES and existing techniques for various numbers of blocks.

## 6. Conclusions

The CC and big data evolved together with the development of information management from basic reporting and querying, advanced analytics, business intelligence and machine learning. The privacy and security of big data is a problematic issue that restricts organizations in utilizing Cloud services. In order to solve such an issue, TDES methodology is proposed to provide security for the big data in Cloud environments by using healthcare data. Initially, the input selection was carried out where the healthcare dataset was considered as an input. The datasets include attributes of patients such as name, gender, age, month, symptoms and location. After selecting the data, input was encrypted by using TDES methodology. The encrypted big health care data are then stored in the Cloud environments, where it superiorly supports read-write operations on storage. To retrieve the data from the Cloud environment, the process of data decryption is utilized by using TDES methodology. The proposed TDES methodology provided a simpler technique by increasing the sizes of keys in DES to protect against the attacks and defend the privacy of data. The experimental results proved that the proposed TDES method is effective in providing security and privacy to big healthcare data in Cloud environment. However, the proposed model requires higher network utilization and CPU usage. Therefore, as a future extension, a modified elliptic curve based cryptographic methodology will be integrated with the block-chain technology to further reduce the CPU usage, network utilization and computational time in the Cloud environment, which are the major concerns described in the literature section. In addition to this, the proposed model is tested on image and audio files to further enhance and ensure security for Cloud-based services, because the multimodal data analysis is effective in precise treatment and early disease diagnosis.

**Author Contributions:** The paper investigation, resources, data curation, writing—original draft preparation, writing—review and editing and visualization were done by M.N.R. and M.S.R. The paper conceptualization and software were conducted by J.A.B. and K.L.H. The validation and formal analysis, methodology, supervision, project administration and funding acquisition of the version to be published were conducted by W.C.L. and B.D.P. All authors have read and agreed to the published version of the manuscript.

**Funding:** This research received no external funding.

**Institutional Review Board Statement:** Ethical review and approval were waived for this study, since this article doesn't involve any clinical research or practices in it.

**Informed Consent Statement:** Informed consent was obtained from all subjects involved in the study.

**Data Availability Statement:** All data generated or analyzed during this study are included in this published article. Publicly available datasets were analyzed in this study. This data can be found here: https://github.com/Parameshachari/Healthcare-Dataset-for-cloud-security#healthcare-dataset-for-cloud-security (accessed on 9 September 2022).

**Conflicts of Interest:** The authors declare no conflict of interest.

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
