# Peer review of "An Efficient and Secure Big Data Storage in Cloud Environment by Using Triple Data Encryption Standard"

_2504-2289, doi:10.3390/bdcc6040101_

Round 1

Reviewer 1 Report

The manuscript is well structured and is easy to read.

Despite the above, the manuscript, in the reviewer's opinion, still has room for improvement.

A more detailed definition of the theoretical framework is lacking. Having in consideration that   NIST has deprecated DES and 3DES for new applications in 2017, and for all applications by 2023, the authors should motivate their work, why is proper in the actual situation to continue the research work regarding 3DES. This might be expected to be developed in the introductory section, before moving on to section number two.

Author Response

Thanks for the constructive comments. The motivation of the research has been provided in the introductory section.

Reviewer 2 Report

In the introduction, sufficient background on the field is presented and relevant references are included.

In section 2, a detailed literature review is presented concerning the relevant references.

Section 3 presents a detailed methodology for TDES algorithm providing security for the bigdata in cloud environment by using healthcare data that is well described.

In section 4, the results of the empirical experiments are presented. Three main Performance Measures were investigated - Execution Time, Network Usage and CPU utilization. The results are clearly presented with tables and diagrams. Comparisons with other methods are made.

In the conclusion, general conclusions are drawn, which are supported by the results of the study.

The research design is appropriate.

As general the paper is interesting and well structured.

Author Response

In the introduction, sufficient background on the field is presented and relevant references are included.

Thanks for your appreciating comments.

In section 2, a detailed literature review is presented concerning the relevant references.

Thanks for your appreciating comments.

Section 3 presents a detailed methodology for TDES algorithm providing security for the bigdata in cloud environment by using healthcare data that is well described.

Thanks for noticing our efforts and providing a valuable comments  .

In section 4, the results of the empirical experiments are presented. Three main Performance Measures were investigated - Execution Time, Network Usage and CPU utilization. The results are clearly presented with tables and diagrams. Comparisons with other methods are made.

Thanks for your valuable feedback.

In the conclusion, general conclusions are drawn, which are supported by the results of the study.

Thanks for your appreciating comments.

The research design is appropriate.

Thanks for the comment.

As general the paper is interesting and well structured.

Thanks for all your valuable feedback.

Reviewer 3 Report

The paper refers to a topical issue, presenting TDES algorithm. While the paper has a potential, there is a list of improvements to be made prior the paper can be published.

"Bigdata" consists of two words, while through the text the authors use this as a single term.

The authors states "To store bigdata in cloud organization requires higher computing power for larger sensitive data and larger space for storing data 20and results" - (1) storage is not so closely related to computing power, while this is more about processing, (2) similarly computing power is less related to "sensitive" data, while you do not speak about additional efforts to be put dealing with sensitive data, while generally computing power is about all types of larger amounts and variety of data.

" to provide security for the bigdata in cloud environment by using 25healthcare data" - make sure that the last part, i.e. reference to healthcare data, is presented as an application domain. Otherwise, the current form may be misinterpreted that the proposal is valid only for them, which is not correct.

In the current digital world, "The Proposed TDES 29method shows lesser execution time of 48 minutes for encryption, storage and decryption" with no additional context doe snot sound impressive, where we mostly speak about processing data on fly, in real-time or nearly real-time. Therefore, I would suggest to revise this statement and not to use 48 minutes in the abstract, where you cannot elaborated on this in more details, being more abstract, while this results are presented later with an in-depth discussion.

Statements such as "The Cloud Computing (CC) is an fastly developed business storage computing plat-38form and it includes many advantages like larger storage, scalability and lower cost." should also have external references used as a kind of evidence / proof.

The role of FIgure 1 is not very clear. On the one hand, it provides a very elementary basic details that could be easily discussed in a text, on the other hand, the Figures in Introduction is something not very traditional / typical. Reconsider the need for this Figure in Introduction.

"The cloud is an emergent technique in the technology of data analytics that is used 54to retrieve and stockup the bigdata in distributed environment." - the authors are invited to be both, more formal and accurate in their statements. I.e. "cloud" is a formal term, especially, when the authors elaborate on the technique and technology, not a storage.

In addition, considering that the authors put an emphasis on the security, there is an expressed need to refer to the evidences from the literature and practice demonstrating the topicality of this issue. As an example, see following references covering security issues of NoSQL DB dealing with Big Data and / or deploying the database in the web for cloud computing purposes:

1.      Daskevics A. et al. (2021, December). IoTSE-based open database vulnerability inspection in three Baltic countries: ShoBEVODSDT sees you. In 2021 8th International Conference on Internet of Things: Systems, Management and Security (IOTSMS) (pp. 1-8). IEEE.

2.      Ferrari D. et al. (2020,December). NoSQL Breakdown: A Large-scale Analysis of Misconfigured NoSQL Services. In Annual Computer Security Applications Conference (pp.567-581)

3.      Nikiforova A. et al. (2022). NoSQL security: can my data-driven decision-making be affected from outside?. arXiv preprint arXiv:2206.11787.

4.      Derbeko P, et al. (2016).Security and privacy aspects in MapReduce on clouds: A survey. Computer science review,20, 1-28.

Some of them can be used as a source for further references (via snowballing approach) covering many important your study-related aspects.

The authors list contributions of the paper, which contributes to the general understanding and improves the structure, while from the content of those points, I would say that only the first numbered bullet is a contribution, while the following points are rather implying and describes the original contribution in more details, emphasizing its different aspects. All in all, they are not equally weighted, which makes the list debatable.

The first sentence of the Section 2 states "The existing techniques on the basis of big data on cloud computing methods are92reviewed in this section", while in the context of this paper, it is crucial to cover (1) the scope of this studies, i.e. what was the context you were interested in, where combination of Big Data and Cloud Computing is not sufficient, considering the wide popularity of combination of this topic, (2) how these studies were selected? Was this a result of SLR? or what approach was used and how could you "prove" that all relevant studies were covered constituting a sufficient knowledge base?

The authors are invited to improve the added-value of the review of literature in the context of the conducted study. Please, make sure you emphasize key takeaways and point out the gap you are filling in.

In section 3 I would suggest to start with a very generalized version of the proposed methodology, which would then present a more specific examples, instead of starting with this example.

"The advanced encryption standard is replaced by Data Encryption 228Standard (DES) so; it is considered as outdated at present." - please, provide the evidence, i.e.external reference.

"There are several types of TDES encryption that are normally recognized in 232three ways that is explained in the following..." with the following elaboration on these types also require references, which (1) supports this information, (2) refers the reader to the sources, where more information on this may be obtained.

"In the three key options to utilize the subkeys first option is best because all the three 265subkeys includes different combination having effective key length of 168 bit. So, it is dif-266ficult to encrypt the data using first option and is difficult to resolve the usage of second 267and third option. " - please, review this part since for me it sounds contradictorary.

Section 3.5 provides metrics and elaborate sin the result within this set, which is a good point. But, at the very same time, the selection of these metrics should be justified and made methodologically and scientifically soundy. It should also be elaborated on why this set constitutes the fullest set and no other metrics should be involved. Again, this can be done via external literature.

Section 3.6 presents comparative results with other two techniques. First, it would be needed to motivate a choice of these two techniques, what I missed or what was not presented. Second, while Execution Time is better, the proposed technique demonstrate worse result in CPU usage and Network Utilization, which should be discussed. This is important in particular in the context "The proposed TDES algorithm provides fairly simpler technique by increasing the sizes of keys in DES to protect against the attacks and defends the privacy of data", i.e. it is simpler, and while demonstrating a bit better Execution time, requires higher network utilization and CPU usage? In addition to the discussion, some suggestions on how this can be overcome and improved are needed.

The authors are strongly encouraged to add Section on Limitations of the study and discuss extensively Future work, as well as how / when the method proposed can / should be used and why? Considering some drawbacks in relation to the metrics they cover.

Also, it would be beneficial to see not only the part of encryption = securing the data, but also decryption and the level/ extent of the vulnerability of both the proposed method and the one to which it is compared to.

The paper also requires extensive improvements in the language, including the grammar, style etc. Involvement of the native speaker would benefit.

All in all, the paper has a potential, but it requires improvements through the paper.

Author Response

The paper refers to a topical issue, presenting TDES algorithm. While the paper has a potential, there is a list of improvements to be made prior the paper can be published.

"Bigdata" consists of two words, while through the text the authors use this as a single term.

Thanks for your valuable suggestions. The term “Bigdata” is replaced with “Big Data” through out the paper.

The authors states "To store bigdata in cloud organization requires higher computing power for larger sensitive data and larger space for storing data 20and results" - (1) storage is not so closely related to computing power, while this is more about processing, (2) similarly computing power is less related to "sensitive" data, while you do not speak about additional efforts to be put dealing with sensitive data, while generally computing power is about all types of larger amounts and variety of data.

Thanks for the comments. The main objective of the research is to provide security and privacy for the health care big data. The storage and computing power are types of data that are least discussed in the research.

" to provide security for the bigdata in cloud environment by using 25healthcare data" - make sure that the last part, i.e. reference to healthcare data, is presented as an application domain. Otherwise, the current form may be misinterpreted that the proposal is valid only for them, which is not correct.
Thanks for the valuable comments. The term ‘health care data’ is present as a big data in the research paper.

In the current digital world, "The Proposed TDES 29method shows lesser execution time of 48 minutes for encryption, storage and decryption" with no additional context does not sound impressive, where we mostly speak about processing data on fly, in real-time or nearly real-time. Therefore, I would suggest to revise this statement and not to use 48 minutes in the abstract, where you cannot elaborate on this in more details, being more abstract, while these results are presented later with an in-depth discussion.

Thanks for the valuable comments. The time term is removed from the abstract.

Statements such as "The Cloud Computing (CC) is an fastly developed business storage computing plat-38form and it includes many advantages like larger storage, scalability and lower cost." should also have external references used as a kind of evidence / proof.

Thanks for the constructive comments. An external reference has been provided as [2] in the introduction section.

The role of Figure 1 is not very clear. On the one hand, it provides a very elementary basic details that could be easily discussed in a text, on the other hand, the Figures in Introduction is something not very traditional / typical. Reconsider the need for this Figure in Introduction.

Thanks for the constructive comments. As figure 1 was a general architecture, it has been removed.

"The cloud is an emergent technique in the technology of data analytics that is used 54to retrieve and stockup the bigdata in distributed environment." - the authors are invited to be both, more formal and accurate in their statements. I.e. "cloud" is a formal term, especially, when the authors elaborate on the technique and technology, not a storage.

Thanks for the constructive comments. The main objective of the research is to provide security and privacy for the health care big data. The storage term is least discussed in the research.

In addition, considering that the authors put an emphasis on the security, there is an expressed need to refer to the evidences from the literature and practice demonstrating the topicality of this issue. As an example, see following references covering security issues of NoSQL DB dealing with Big Data and / or deploying the database in the web for cloud computing purposes:

  1. Daskevics A. et al. (2021, December). IoTSE-based open database vulnerability inspection in three Baltic countries: ShoBEVODSDT sees you. In 2021 8th International Conference on Internet of Things: Systems, Management and Security (IOTSMS)(pp. 1-8). IEEE.
  2. Ferrari D. et al. (2020,December). NoSQL Breakdown: A Large-scale Analysis of Misconfigured NoSQL Services. In Annual Computer Security Applications Conference (pp.567-581)
  3. Nikiforova A. et al. (2022). NoSQL security: can my data-driven decision-making be affected from outside?. arXiv preprint arXiv:2206.11787.
  4. Derbeko P, et al. (2016).Security and privacy aspects in MapReduce on clouds: A survey. Computer science review,20, 1-28.

Some of them can be used as a source for further references (via snowballing approach) covering many important your study-related aspects.

Thanks for the constructive comments. The aforementioned sources have been referred in the introduction section [5-8].

The authors list contributions of the paper, which contributes to the general understanding and improves the structure, while from the content of those points, I would say that only the first numbered bullet is a contribution, while the following points are rather implying and describes the original contribution in more details, emphasizing its different aspects. All in all, they are not equally weighted, which makes the list debatable. 

Thanks for the constructive comments. The contributions and objectives of the present research work is provided in the introduction section.

The first sentence of the Section 2 states "The existing techniques on the basis of big data on cloud computing methods are92reviewed in this section", while in the context of this paper, it is crucial to cover (1) the scope of this studies, i.e. what was the context you were interested in, where combination of Big Data and Cloud Computing is not sufficient, considering the wide popularity of combination of this topic, (2) how these studies were selected? Was this a result of SLR? or what approach was used and how could you "prove" that all relevant studies were covered constituting a sufficient knowledge base?

Thanks for the valuable comments. The scope of the research has been provided and the importance of selecting those particular studies in the literature section has also been provided in the section 2 as “Computing infrastructure, especially cloud computing, plays a significant role …. transactions are digitally recorded”.

The authors are invited to improve the added-value of the review of literature in the context of the conducted study. Please, make sure you emphasize key takeaways and point out the gap you are filling in.

Thank you for constructive comments. The importance of selecting those particular studies in the literature section has also been provided and the research gap has also been identified at last in the section 2. It is provided as “Some of the attributes-based encryption technique…. the computation time of model”.

 In section 3 I would suggest to start with a very generalized version of the proposed methodology, which would then present a more specific examples, instead of starting with this example. 

Thanks for the valuable suggestions. The generalized version of the proposed methodology is given in section 3.1.

"The advanced encryption standard is replaced by Data Encryption 228Standard (DES) so; it is considered as outdated at present." - please, provide the evidence, i.e.external reference.
Thank you for constructive comments. The external reference as [25] is provided as evidence.

"There are several types of TDES encryption that are normally recognized in 232three ways that is explained in the following..." with the following elaboration on these types also require references, which (1) supports this information, (2) refers the reader to the sources, where more information on this may be obtained.

Thanks for the valuable comments. The external reference as [25] is provided as evidence.

"In the three key options to utilize the subkeys first option is best because all the three subkeys includes different combination having effective key length of 168 bit. So, it is difficult to encrypt the data using first option and is difficult to resolve the usage of second and third option. " - please, review this part since for me it sounds contradictory. 

Thanks for the valuable suggestions. As the encryption by first option was best there was no need to undergo for the second and third option.

Section 3.5 provides metrics and elaborate sin the result within this set, which is a good point. But, at the very same time, the selection of these metrics should be justified and made methodologically and scientifically soundy. It should also be elaborated on why this set constitutes the fullest set and no other metrics should be involved. Again, this can be done via external literature.

Thanks for the valuable comments. The selection of the metrics is done by comparing the existing research works. Thus, common metrics are evaluated in the proposed research work. 

Section 3.6 presents comparative results with other two techniques. First, it would be needed to motivate a choice of these two techniques, what I missed or what was not presented. Second, while Execution Time is better, the proposed technique demonstrate worse result in CPU usage and Network Utilization, which should be discussed. This is important in particular in the context "The proposed TDES algorithm provides fairly simpler technique by increasing the sizes of keys in DES to protect against the attacks and defends the privacy of data", i.e. it is simpler, and while demonstrating a bit better Execution time, requires higher network utilization and CPU usage? In addition to the discussion, some suggestions on how this can be overcome and improved are needed.

Thank you for constructive comments. Yes, as per your comment, the actual CPU utilization varied on various machines but the proposed model showed better values than existing models. This is because, during computing the tasks, the model requires higher network utilization and CPU usage. The problem can be overcome by Enabling High-Performance Mode to enhance the performances of the system.  

The authors are strongly encouraged to add Section on Limitations of the study and discuss extensively Future work, as well as how / when the method proposed can / should be used and why? Considering some drawbacks in relation to the metrics they cover.

Thank you for constructive comments. The proposed model requires higher network utilization and CPU usage thus, in the future, choosing the right cloud security provider can enhance and ensure sound security for cloud-based services.

Also, it would be beneficial to see not only the part of encryption = securing the data, but also decryption and the level/ extent of the vulnerability of both the proposed method and the one to which it is compared to.

Thank you for constructive comments. The research objective is to encryption, we have incorporated more about encryption. As per your recommendation, in future we will incorporate about decryption also.  

The paper also requires extensive improvements in the language, including the grammar, style etc. Involvement of the native speaker would benefit.

Thank you for constructive comments. We have improved the language in the research work.

Round 2

Reviewer 3 Report

The authors made an iteration to improve the quality of the paper in line with the comments received. Some of the above mentioned concerns were resolved successfully. However, there is a list that remains valid, as well as new comments come with the changes made in the paper.

There are sentences that must be reworded as being either inaccurate or incorrect. E.g. "The Big Data in cloud organization is for providing security for larger sensitive data and results." - ensuring security of sensitive data is not the primary goal of the Big Data, which makes this sentence very inaccurate, which is even more so considering the later sentence, where it is stated that for this purpose an algorithm will be presented in the paper.

In "The Proposed TDES method showedlesser execution time for encryption, storage and decryption, whereas the existing IFHDS method showed execution time of 53" - following my previous comment, the authors removed the results of their algorithm, while left the AVG of existing methods, which make this sentence very inconsistent. Either remove both results, stating that your is better compared with the existing in this respect, or leave as it was mentioning both - but the consistency should be ensured.

"The cloud is an emergent technique" - "cloud" is too formal (cloud computing, cloud storage etc. is the one), and moreover, cloud is not a technique. Either specify what is the cloud artifact you are speaking about OR replace the term "technique" with appropriate term.

"The cloud is an emergent technique in the technology of data analytics that is used 52to retrieve Big Datain distributed environment" - retrieval of the data is not the only purpose of using cloud technologies. Please reword to make the statement accurate.

" Handling 68privacy is both a technical and a sociological problem" - "sociological" is not a very popular and widely used term. Reword or elaborate on it. Actually "sociotechnical" is the one, which should be used here instead of two you refer to.

"thesefactors havemotivated to improve the security for big 70data in cloud environment." - the paper contributes to the improving security but does not improve it at a high level, i.e. such strong formulation as you use require significantly more general, comprehensive and universal improvement of the security for all big data domain. Please, reword to make it more paper-compliant.

"The objective and contribution of this research is dis-73cussed as below" - objectives and contribution should be splitted. Moreover, all bullets must be made equally important, i.e. the 1st sentence of the 1st bullet is an objective, while the second although is expected to be such, is not, being rather a continuation of the first and most probably should be presented as a contribution, i.e. continuation of the contribution. The list supposes that all bullet points will be of the same level of importance. Please, ensure this.

"TDES algorithm is proposed to provide security for Big Datain the 196cloud environment,by using healthcare data" - probably you want to say that the healthcare data are used as an application domain.

"it supports input and output operations" should be elaborated in more details, i.e. is this only a reference to read-write operations? It is also recommended to emphasize that updates are not expected here (if applicable).

"enablestorage with various data objects and 204files." - please elaborate on these data objects and files. Are there any limitations? If no, just discuss this in the context of what Big Data objects usually tend to be AND more importantly, what was the data objects and files you used in your experiment. The latter is of very high importance.

The data you used in the experiment should be discussed in detail. Provide a short example on how they looked like. What were their structure? Amount? What types of Big Data specific objects and specificities are present there? Which of them are not in place? And for those not presented in your dataset - how you can proof that the algorithm will work equally with them? I.e. if images highly relevant for the healthcare data would affect the results? Would the algorithm still act faster compared to alternatives? How those results, which are worse compared to existing would be affected? Their level of accessibility and openness should also be mentioned, elaborating on why the open data were not used (if it is the case).

"The proposed model requires higher network utilization and CPU usage thus, in the future, 534choosing the right cloud security provider can enhance & ensure sound security for cloud-535based services." this should be discussed in detail. I.e. not only the fact of such, but also speculations and assumptions on how this could be achieved?

It is crucial to cover (1) the scope of this studies, i.e. what was the context you were interested in, where combination of Big Data and Cloud Computing is not sufficient, considering the wide popularity of combination of this topic, (2) how these studies were selected? Was this a result of SLR? or what approach was used and how could you "prove" that all relevant studies were covered constituting a sufficient knowledge base?

very many statements still require an external reference to be used as sort of "evidence" making them less subjective.

The authors are strongly encouraged to add Section on Limitations of the study and discuss extensively Future work, as well as how / when the method proposed can / should be used and why? Considering some drawbacks in relation to the metrics they cover.

To sum up, at least the above comments require an attention. It is crucial to pay a very serious attention to the comments related to the types of data you covered and how these results will be affected if others data types relevant for Big Data and healthcare domain in particular will be involved - i.e. your dataset is not sufficiently discussed AND does not include media, including but not limited to images, which are highly relevant for healthcare domain - how the results would be affected, if they would be in place? Since this could require a significant amount of the additional work, one of the easiest options would be to adapt the title, abstract, intro and conclusions stating that a very simple example of mostly textual data will be considered.

Author Response

  1. There are sentences that must be reworded as being either inaccurate or incorrect. E.g. "The Big Data in cloud organization is for providing security for larger sensitive data and results." - ensuring security of sensitive data is not the primary goal of the Big Data, which makes this sentence very inaccurate, which is even more so considering the later sentence, where it is stated that for this purpose an algorithm will be presented in the paper.

Thank you for your valuable comments. The following sentence is modified as “In recent decades, big data has become the most important technology for handling, collecting, and analyzing the huge amount of unstructured, structured, and semi-structured data in a higher performance environment. Hence, big data security offers cloud application security and monitoring to host highly sensitive data to support cloud platforms. However, the privacy and security of big data has become an emerging issue that restricts the organization to utilize cloud services” in the abstract section.

  1. In "The Proposed TDES method showed lesser execution time for encryption, storage and decryption, whereas the existing IFHDS method showed execution time of 53" - following my previous comment, the authors removed the results of their algorithm, while left the AVG of existing methods, which make this sentence very inconsistent. Either remove both results, stating that your is better compared with the existing in this respect, or leave as it was mentioning both - but the consistency should be ensured.

Thank you for your valuable comments. The respective sentence is modified as “The experimental results show that the proposed TDEF method is effective in providing security and privacy to big healthcare data in the Cloud environment. The proposed TDES method showed lesser encryption and decryption time compared to the existing IFHDS method”.

  1. "The cloud is an emergent technique" - "cloud" is too formal (cloud computing, cloud storage etc. is the one), and moreover, cloud is not a technique. Either specify what is the cloud artifact you are speaking about OR replace the term "technique" with appropriate term.

Thank you for your valuable comments. The respective sentence is updated in the introduction section as “Cloud computing is an emergent technology in data analytics, which is used to retrieve, store and share Big Data in a distributed environment.”

  1. "The cloud is an emergent technique in the technology of data analytics that is used to retrieve Big Data in distributed environment" - retrieval of the data is not the only purpose of using cloud technologies. Please reword to make the statement accurate.

Thank you for your valuable comments. The respective sentence is updated in the introduction section as “Cloud computing is an emergent technology in data analytics, which is used to retrieve, store and share Big Data in a distributed environment.”

  1. " Handling privacy is both a technical and a sociological problem" - "sociological" is not a very popular and widely used term. Reword or elaborate on it. Actually "sociotechnical" is the one, which should be used here instead of two you refer to.

Thank you for your valuable comments.

Dear reviewer, in the introduction section, the sentence is updated as “Handling privacy is a socio-technical problem, which must be realized to take advantage of Big Data”.

  1. " These factors have motivated to improve the security for big data in cloud environment." - the paper contributes to the improving security but does not improve it at a high level, i.e. such strong formulation as you use require significantly more general, comprehensive and universal improvement of the security for all big data domain. Please, reword to make it more paper-compliant.

Thank you for your valuable comments. We updated the respective sentence as “Handling privacy is a socio-technical problem, which must be realized to take advantage of Big Data. To overcome such an issue, Triple Data Encryption Standard (TDES) algorithm is proposed in this article to further enhance security in the cloud environment, especially related to healthcare applications”.

  1. "The objective and contribution of this research is discussed as below" - objectives and contribution should be splitted. Moreover, all bullets must be made equally important, i.e. the 1st sentence of the 1st bullet is an objective, while the second although is expected to be such, is not, being rather a continuation of the first and most probably should be presented as a contribution, i.e. continuation of the contribution. The list supposes that all bullet points will be of the same level of importance. Please, ensure this.

Thank you for your valuable comments. The contributions of this research are made equally important and maintain the same level of importance.

  1. "TDES algorithm is proposed to provide security for Big Data in the 196 cloud environment by using healthcare data" - probably you want to say that the healthcare data are used as an application domain.

Thank you for your valuable comments.

Dear reviewer, we updated the respective sentence as “To overcome such an issue, Triple Data Encryption Standard (TDES) algorithm is proposed in this article to further enhance security in the cloud environment, especially related to healthcare applications.”

  1. "It supports input and output operations" should be elaborated in more details, i.e. is this only a reference to read-write operations? It is also recommended to emphasize that updates are not expected here (if applicable).

Thank you for your valuable comments.

Dear reviewer, the respective sentence is modified as follows “The TDES is a standard open encryption technique, which provides key strength of 112 bit and 168 bit for encryption. The encrypted healthcare-based big data is stored in the cloud environments. Once the encryption process is completed, the next process: decryption is performed to retrieve the data from the cloud environments. The process of data decryption is accomplished by using TDES methodology.”

  1. "Enable storage with various data objects and files." - Please elaborate on these data objects and files. Are there any limitations? If no, just discuss this in the context of what Big Data objects usually tend to be AND more importantly, what was the data objects and files you used in your experiment. The latter is of very high importance.

Thank you for your valuable comments. The following sentence is updated in the conclusion and section 3.3 as “The encrypted big health care data are stored in the cloud environments, where it effectively supports read-write operations on storage [25, 26].”

  1. The data you used in the experiment should be discussed in detail. Provide a short example on how they looked like. What were their structure? Amount? What types of Big Data specific objects and specificities are present there? Which of them are not in place? And for those not presented in your dataset - how you can proof that the algorithm will work equally with them? I.e. if images highly relevant for the healthcare data would affect the results? Would the algorithm still act faster compared to alternatives? How those results, which are worse compared to existing would be affected? Their level of accessibility and openness should also be mentioned, elaborating on why the open data were not used (if it is the case).

Thank you for your valuable comments.

Dear reviewer, as mentioned in section 3.1, the real time healthcare dataset is used for experimental investigation. The collected dataset includes 17 attributes such as patient name, gender, age, month, symptoms, location, etc., and 3024 instances. The respective collected dataset will be online, once the paper gets published. Additionally, we have not performed experiments on the image datasets.

  1. "The proposed model requires higher network utilization and CPU usage thus, in the future, choosing the right cloud security provider can enhance & ensure sound security for cloud based services." this should be discussed in detail. I.e. not only the fact of such, but also speculations and assumptions on how this could be achieved?

Thank you for your valuable comments.

Dear reviewer, the future work is elaborated as “However, the proposed model requires higher network utilization and CPU usage. Therefore, as the future extension, an effective deep learning based encryption methodology can be developed and an appropriate cloud security provider can be selected for enhancing and ensuring security for cloud-based services”

  1. The authors are strongly encouraged to add Section on Limitations of the study and discuss extensively Future work, as well as how / when the method proposed can / should be used and why? Considering some drawbacks in relation to the metrics they cover.

Thank you for your valuable comments.

Dear reviewer, the limitation of each literature is given in section 2. Additionally, the relationship between the existing literature and the proposed model is given as a separate paragraph at the end of section 2. In addition, the future work is elaborated in the current manuscript.

  1. It is crucial to cover (1) the scope of this studies, i.e. what was the context you were interested in, where combination of Big Data and Cloud Computing is not sufficient, considering the wide popularity of combination of this topic, (2) how these studies were selected? Was this a result of SLR? or what approach was used and how could you "prove" that all relevant studies were covered constituting a sufficient knowledge base? 

Thank you for your valuable comments.

Dear reviewer, as depicted in the updated manuscript. “After dataset acquisition, the data selection is processed, in which the encryption of these input data is done by using TDES methodology. The encrypted healthcare-based big data is stored in the cloud environments. Once the encryption process is completed, the next process: decryption is performed to retrieve the data from the cloud environments. The process of data decryption is accomplished by using TDES methodology.” In this manuscript, the external cloud security is improved by implementing TDES methodology.

  1. Very many statements still require an external reference to be used as sort of "evidence" making them less subjective.

Thank you for your valuable comments. Three more references (33rd, 34th and 35th papers) are referred for external evidence.

  1. To sum up, at least the above comments require an attention. It is crucial to pay a very serious attention to the comments related to the types of data you covered and how these results will be affected if others data types relevant for Big Data and healthcare domain in particular will be involved - i.e. your dataset is not sufficiently discussed AND does not include media, including but not limited to images, which are highly relevant for healthcare domain - how the results would be affected, if they would be in place? Since this could require a significant amount of the additional work, one of the easiest options would be to adapt the title, abstract, intro and conclusions stating that a very simple example of mostly textual data will be considered.

Thank you for your valuable comments. The above mentioned comments are clearly updated in the current manuscript

Round 3

Reviewer 3 Report

The authors made another iteration improving the paper. However, it still has the list of issues to be resolved, containing those unresolved from previous iterations.

In more detail:

" big data has become the most important technology" - big data is not a technology, i.e., data != technology.

Similarly, "big data has become the most important technology for handling, collecting,and analyzing " - big data being data cannot be addressed as something that collect, analyze etc. the data, i.e., this sentence can be simplified as "data is a technology that handle, collect and analyze data", which is obviously inaccurate. Please, be very careful in selecting terms and definitions for the terms considered since this affects the perception of the quality of the whole idea.

"Once the encryptionprocessis completed, the next process:decryp-211tionis performed to retrieve the data from the cloud environments. " is also not very accurate since the decryption is only needed once the data are requested, i.e., not immediately after they have been encrypted. Thus, the rewording is needed.

"the real time healthcare 222dataset is used for experimental investigation. The collected dataset " - the "collected" refers more to stand-alone data, while you deal with real-time data, which rather requires "retrieved".

The authors are also invited to decide on and use consistently the term associated with the TDES, i.e., do not mix methodology and algorithm, which are semantically different concepts.

The resolution of Figures is insufficient. The content of Figure 1 is difficult to read, while Figure 2 is completely unreadable.

Please elaborate on "data objects" and "files" the algorithm supports, was tested on and is supposed for. Are there any limitations? Discuss this in the context of what Big Data objects usually tend to be AND more importantly, what was the data objects and files you used in your experiment? Do not ignore images and audio files that play a very important role in the healthcare domain. The latter is of very high importance.

Please, decide on and use consistently, is TDES methodology or algorithm as you mix both of them, while they are semantically different objects.

The comment you received before "It is crucial to cover (1) the scope of this studies, i.e. what was the context you were interested in, where combination of Big Data and Cloud Computing is not sufficient, considering the wide popularity of combination of this topic, (2) how these studies were selected? Was this a result of SLR? or what approach was used and how could you "prove" that all relevant studies were covered constituting a sufficient knowledge base? " refers to the literature review you provide, not the proposed algorithm.

"The authors are strongly encouraged to add Section on Limitations of the study and discuss extensively Future work, as well as how / when the method proposed can / should be used and why? Considering some drawbacks in relation to the metrics they cover." refers to the current study not the existing literature.

Previous comment "The proposed model requires higher network utilization and CPU usage thus, in the future, choosing the right cloud security provider can enhance & ensure sound security for cloud based services." this should be discussed in detail. I.e. not only the fact of such, but also speculations and assumptions on how this could be achieved?" expected an elaboration on this in detail, while the piece of text available in the current version is just a mention of such. Please, elaborate on this in detail.

The data you used in the experiment should be discussed in detail. Provide a short example on how they looked like. What were their structure? Amount? What types of Big Data specific objects and specificities are present there? For this purpose provide a very short example listing all attributes and 1-2 sets of values. Are they only textual and numerical? And for those Big Data specific data types not presented in your dataset - how you can proof that the algorithm will work equally with them? I.e. if images highly relevant for the healthcare data would affect the results? Would the algorithm still act faster compared to alternatives? How those results, which are worse compared to existing would be affected?

In addition, the authors state that the dataset will become available once the paper is accepted - respective statement and later the link should be added to the text. In addition, considering that the authors say they dealt with real-time data, it means that rather a snapshot will provided, which should be also mentioned.

It is still very important to pay an attention to the comments related to the types of data you covered and how these results will be affected if others data types relevant for Big Data and healthcare domain in particular will be involved - i.e. your dataset is not sufficiently discussed AND does not include media, including but not limited to images, which are highly relevant for healthcare domain - how the results would be affected, if they would be in place? Since this could require a significant amount of the additional work, one of the easiest options would be to adapt the title, abstract, intro and conclusions stating that a very simple example of mostly textual data will be considered.

Author Response

  1. " Big data has become the most important technology" - big data is not a technology, i.e., data! = technology.

Thank you for your valuable comments. The corresponding sentence is replaced as follows “In recent decades, big data analysis has become the most important research topic. Hence, big data security offers cloud application security and monitoring to host highly sensitive data to support cloud platforms. However, the privacy and security of big data has become an emerging issue that restricts the organization to utilize cloud services.”

  1. Similarly, "big data has become the most important technology for handling, collecting, and analyzing " - big data being data cannot be addressed as something that collect, analyze etc. the data, i.e., this sentence can be simplified as "data is a technology that handle, collect and analyze data", which is obviously inaccurate. Please, be very careful in selecting terms and definitions for the terms considered since this affects the perception of the quality of the whole idea.

Thank you for your valuable comments. The respective sentence is updated as follows “In recent decades, big data analysis has become the most important research topic. Hence, big data security offers cloud application security and monitoring to host highly sensitive data to support cloud platforms. However, the privacy and security of big data has become an emerging issue that restricts the organization to utilize cloud services.”

  1. "Once the encryption process is completed, the next process: decryption is performed to retrieve the data from the cloud environments. " is also not very accurate since the decryption is only needed once the data are requested, i.e., not immediately after they have been encrypted. Thus, the rewording is needed.

Thank you for your valuable comments. The following sentence is updated as follows in the methodology section, “Once the encryption process is completed, after requesting a specific data, the decryption process is performed on the corresponding requested data in order to retrieve the data from the cloud environments”.

  1. "The real time healthcare dataset is used for experimental investigation. The collected dataset”- the "collected" refers more to stand-alone data, while you deal with real-time data, which rather requires "retrieved"

Thank you for your valuable comments. The word “collected” is replaced as “retrieved” in the methodology section, especially the content related to the dataset description.

  1. The authors are also invited to decide on and use consistently the term associated with the TDES, i.e., do not mix methodology and algorithm, which are semantically different concepts.

Thank you for your valuable comments. We have followed TEDS as a methodology not as an algorithm.

  1. The resolution of Figures is insufficient. The content of Figure 1 is difficult to read, while Figure 2 is completely unreadable.

Thank you for your valuable comments. The figures 1 and 2 are modified in the updated manuscript.

  1. Please elaborate on "data objects" and "files" the algorithm supports, was tested on and is supposed for. Are there any limitations? Discuss this in the context of what Big Data objects usually tend to be AND more importantly, what was the data objects and files you used in your experiment? Do not ignore images and audio files that play a very important role in the healthcare domain. The latter is of very high importance.

Thank you for your valuable comments. The data we used in the experiment is discussed effectively in the methodology section with their structure and amount. As specified in the methodology section, the real time healthcare dataset is used for experimental investigation. The dataset includes 17 attributes such as patient name, gender, age, month, symptoms, location, disease, resting blood pressure, resting electrocardiographic results, serum cholesterol, maximum heart rate achieved, past history, consultant name, body mass index, body weight, height, and job type and has 3024 instances. From the patients attributes, the required data are selected and considered for the process of encryption. Before performing encryption, the administrator creates a mask for the personal information attributes such as patient name, gender, age, month, location, past history, consultant name, job type, and body weight. The example data with personal information attributes and key attributes are mentioned in tables 1 and 2. As a future extension, the proposed model can be tested on the image and audio files to further enhance and ensure security for cloud-based services. This statement is clearly mentioned in the conclusion section.

  1. Please, decide on and use consistently, is TDES methodology or algorithm as you mix both of them, while they are semantically different objects.

Thank you for your valuable comments. We have followed TEDS as a methodology not as an algorithm throughout the research manuscript.

  1. The comment you received before "It is crucial to cover (1) the scope of this studies, i.e. what was the context you were interested in, where combination of Big Data and Cloud Computing is not sufficient, considering the wide popularity of combination of this topic, (2) how these studies were selected? Was this a result of SLR? or what approach was used and how could you "prove" that all relevant studies were covered constituting a sufficient knowledge base? “refers to the literature review you provide, not the proposed algorithm.

Thank you for your valuable comments. By reviewing the existing literature studies it has been found that, most of the attributes-based encryption techniques increase the computation time, because they require encryption for attributes as well as data. In addition to this, existing techniques do not maintain proper structure of data that leads to higher computation time. Therefore, a new encryption methodology: TDES is proposed in this manuscript. Whereas, the scope and the motivation of the study is clearly described in section 3.

10."The authors are strongly encouraged to add Section on Limitations of the study and discuss extensively Future work, as well as how / when the method proposed can / should be used and why? Considering some drawbacks in relation to the metrics they cover." refers to the current study not the existing literature.

Thank you for your valuable comments. A separate section is included for the problem statement, which clearly details about the relationship between proposed and the existing methodologies.

  1. Previous comment "The proposed model requires higher network utilization and CPU usage thus, in the future, choosing the right cloud security provider can enhance & ensure sound security for cloud based services." this should be discussed in detail. I.e. not only the fact of such, but also speculations and assumptions on how this could be achieved?" expected an elaboration on this in detail, while the piece of text available in the current version is just a mention of such. Please, elaborate on this in detail.

Thank you for your valuable comments. The future extension is elaborated as follows in the conclusion section “However, the proposed model requires higher network utilization and CPU usage. Therefore, as the future extension, a modified elliptic curve based cryptographic methodology is integrated with the block-chain technology for reducing the CPU usage and network utilization in the cloud environments. In addition to this, the proposed model can be tested on the image and audio files to further enhance and ensure security for cloud-based services.”

  1. The data you used in the experiment should be discussed in detail. Provide a short example on how they looked like. What were their structure? Amount? What types of Big Data specific objects and specificities are present there? For this purpose provide a very short example listing all attributes and 1-2 sets of values. Are they only textual and numerical? And for those Big Data specific data types not presented in your dataset - how you can proof that the algorithm will work equally with them? I.e. if images highly relevant for the healthcare data would affect the results? Would the algorithm still act faster compared to alternatives? How those results, which are worse compared to existing would be affected?

Thank you for your valuable comments. In this manuscript, the real time healthcare dataset is used for experimental investigation. The dataset includes 17 attributes such as patient name, gender, age, month, symptoms, location, disease, resting blood pressure, resting electrocardiographic results, serum cholesterol, maximum heart rate achieved, past history, consultant name, body mass index, body weight, height, and job type and has 3024 instances. From the patients attributes, the required data are selected and considered for the process of encryption. Before performing encryption, the administrator creates a mask for the personal information attributes such as patient name, gender, age, month, location, past history, consultant name, job type, and body weight. The example data with personal information attributes and key attributes are mentioned in tables 1 and 2. As a future extension, the proposed model can be tested on the image and audio files to further enhance and ensure security for cloud-based services. This statement is clearly mentioned in the conclusion section.   

  1. In addition, the authors state that the dataset will become available once the paper is accepted - respective statement and later the link should be added to the text. In addition, considering that the authors say they dealt with real-time data, it means that rather a snapshot will provided, which should be also mentioned.

Thank you for your valuable comments. Tables 1 and 2 are mentioned for the real time healthcare data.

  1. It is still very important to pay an attention to the comments related to the types of data you covered and how these results will be affected if others data types relevant for Big Data and healthcare domain in particular will be involved - i.e. your dataset is not sufficiently discussed AND does not include media, including but not limited to images, which are highly relevant for healthcare domain - how the results would be affected, if they would be in place? Since this could require a significant amount of the additional work, one of the easiest options would be to adapt the title, abstract, intro and conclusions stating that a very simple example of mostly textual data will be considered.

Thank you for your valuable comments. The data we used in the experiment is discussed effectively in the methodology section with their structure and amount, which is specified clearly in tables 1 and 2. As a future extension, the proposed model can be tested on the image and audio files to further enhance and ensure security for cloud-based services. This statement is clearly mentioned in the conclusion section.

Round 4

Reviewer 3 Report

The authors made another round of improvements, where most comments were addressed.

While there are several comments still not addressed, which will be listed below, the paper can be of interest for the audience and now provide details to understand and use it in the future.

For the comments ignored and parts not covered in the manuscript: paper lacks the section devoted to Limitations, which must be clearly indicated especially considering the number of limitations by which the current study can be characterized (mostly referring to the variety of objects that must be covered, i.e., audio, images etc. and how their presence would affect the results). Paper lacks the Section Future works, which should be linked with some of the limitations elaborating in details on how they can be resolved. The scope of the literature reviewed in the theoretical part of the manuscript and the method used for their selection is not presented.

Otherwise, while the above comments would allow to significantly improve the overall quality of the study making it very mature, the paper can be accepted.

Author Response

  1. For the comments ignored and parts not covered in the manuscript: paper lacks the section devoted to Limitations, which must be clearly indicated especially considering the number of limitations by which the current study can be characterized (mostly referring to the variety of objects that must be covered, i.e., audio, images etc. and how their presence would affect the results).

Thank you for your valuable comments. The drawback of each literature is highlighted in section 2. The main motivation of this study and the proposed methodology details are clearly updated in section 3. Majorly, the papers related to the medical field are considered in the literature section. As depicted in the previous round of improvement, as a future extension, the proposed model is tested on the image and audio files to further enhance and ensure security for cloud-based services, because the multimodal data analysis is effective in precise treatment and early disease diagnosis. This statement is clearly mentioned in the conclusion section.

  1. Paper lacks the Section Future works, which should be linked with some of the limitations elaborating in details on how they can be resolved.

Thank you for your valuable comments. The future work is updated as follows “Therefore, as a future extension, a modified elliptic curve based cryptographic methodology is integrated with the block-chain technology for further reducing the CPU usage, network utilization, computational time in the cloud environments, which are the major concerns described in the literature section. In addition to this, the proposed model is tested on the image and audio files to further enhance and ensure security for cloud-based services, because the multimodal data analysis is effective in precise treatment and early disease diagnosis.”

  1. The scope of the literature reviewed in the theoretical part of the manuscript and the method used for their selection is not presented.

Thank you for your valuable comments. As mentioned in the section 3 “By reviewing the existing literature studies, most of the encryption techniques increase the computation time, because they require encryption for large scale attributes and data. The IFHDS, AES, genetic algorithm, CryptDICE, SB-DS, ICE, and Tabu search techniques have large key lengths, which is due to the optimization technique getting trapped into local optima. The existing techniques do not maintain proper structure of data that leads to higher computation time. To address this concern, a new encryption methodology: TDES is proposed in this manuscript. The main motivation of this study is to provide efficient and secure big data storage in a cloud environment with limited computational time.”
